# A Novel Deep-Learning Model Compression Based on Filter-Stripe Group Pruning and Its IoT Application

**DOI:** 10.3390/s22155623

**Published:** 2022-07-27

**Authors:** Ming Zhao, Xindi Tong, Weixian Wu, Zhen Wang, Bingxue Zhou, Xiaodan Huang

**Affiliations:** 1School of Computer Science, Yangtze University, Jingzhou 434023, China; hitmzhao@gmail.com (M.Z.); wx198426@163.com (W.W.); 1164050018wangzhen@gmail.com (Z.W.); 010105zzing@gmail.com (B.Z.); hxd_19795@163.com (X.H.); 2Department of Mathematics and Information Engineering, The Chinese University of Hong Kong, Hong Kong 999077, China

**Keywords:** deep learning, neural network, pruning, target detection, weight quantification

## Abstract

Nowadays, there is a tradeoff between the deep-learning module-compression ratio and the module accuracy. In this paper, a strategy for refining the pruning quantification and weights based on neural network filters is proposed. Firstly, filters in the neural network were refined into strip-like filter strips. Then, the evaluation of the filter strips was used to refine the partial importance of the filter, cut off the unimportant filter strips and reorganize the remaining filter strips. Finally, the training of the neural network after recombination was quantified to further compress the computational amount of the neural network. The results show that the method can significantly reduce the computational effort of the neural network and compress the number of parameters in the model. Based on experimental results on ResNet56, this method can reduce the number of parameters to 1/4 and the amount of calculation to 1/5, and the loss of model accuracy is only 0.01. On VGG16, the number of parameters is reduced to 1/14, the amount of calculation is reduced to 1/3, and the accuracy loss is 0.5%.

## 1. Introduction

Deep neural networks (DNNs) have made significant advances in many fields including speech recognition, computer vision, and natural language processing. However, model deployment is sometimes costly due to the large number of parameters in the DNN. To address this problem, many methods [1,2,3,4,5,6] have been proposed to compress networks and reduce computational quantities. These methods are mainly divided into two categories: structured pruning and unstructured pruning, in which the main method of structured pruning is the filter pruning, while the unstructured pruning method is mainly achieved by weight pruning.

Compared with filter pruning methods, weight pruning is more refined. Weight pruning is mainly implemented by pruning various weights in the network. When the value of a certain weight within the network layer is close to or equal to zero, it can be considered that the prediction performance will not be sacrificed by removing this weight, and hence it can be pruned to eventually form a sparse network. As a result, this method has a drawback in that the positions of the non-zero weights are irregular and random, and we need to additionally record the weight positions. Due to the randomness in the network, the weight pruning method cannot be structured like filter pruning. The speedup cannot be achieved on a universal processor and can only compress the model file size. In contrast, the filter-pruning-based approach performs filter channel pruning at the convolutional layer. Therefore, the pruned network structure is still well-structured and acceleration is easily achieved in a general processor.

The filter pruning process is roughly as follows: (1) Training the larger model until convergence; (2) Splitting the filters according to some criteria; (3) Fine-tuning the network after the construction. It further shows that the pruned model trained with random initialization also achieves high performance. Therefore, it is the structure of the network that matters, not the trained weights. Meanwhile, not only is the structure of the network significant, but also the structure of the filter itself. This can also be seen from the two network structural properties of Figure 1.

For a given output feature, in references [7,8], filters with different kernel sizes (e.g., 1 × 1, 3 × 3, and 5 × 5) were used to convolve and connect all output feature mappings. However, the kernel size of each filter was set manually. Professional experience and knowledge are required to design an efficient network structure, thus the optimal kernel size for each filter can be learned by pruning. Meng et al. [9] proposed the concept of filter strips and found that some stripes have very low $\ell_{1}$-norm values, indicating that such stripes can be removed from the network. The pruning based on the filter strip can better prune the structure of the filter and does not destroy the structure of the whole filter. Sakai et al. [10] performed structured pruning by adaptively deriving the pruning rate for each layer based on the gradient and loss function. Zhang et al. [11] proposed a progressive, multi-step weight pruning framework, as well as network purification and unused path-removal procedures to achieve higher pruning rates without the loss of accuracy. However, the above methods are all based on structured pruning that processes deep neural networks and does not compress the model using up-to-data quantification methods.

Based on this, a refined pruning and weight quantization method based on neural network filters is proposed. The filters in the neural network are firstly refined into stripe-like filter strips. Second, the evaluation of the filter strips is used to refine the partial importance of the filter, cut off the unimportant filter strips and reorganize the remaining filter strips. Finally, of quantification processing is inserted into the training of the neural network after recombination to further compress the computational amount of the neural network. The experimental results show that the proposed method can significantly reduce the computation of neural networks and compress the number of parameters in the model. Based on the experimental results, this method can reduce the number of parameters to 1/4 and the amount of calculation to 1/5, and the loss of model accuracy is only 0.01. On VGG16, the number of parameters is reduced to 1/14, the amount of calculation is reduced to 1/3, and the accuracy loss is 0.5%.

## 2. Related Works

Most of the filter pruning methods of deep neural networks can be divided into three categories: for the entire-filter pruning methods, for the filter channels, and for the filter channel groups. Moreover, there is the weight-based pruning method. The weight-based pruning approach is mainly based on a series of relevant weights in deep neural networks.

There are various pruning criteria in weight pruning. Han et al. [12] pruned network weights based on the $\ell_{1}$-norm criterion and retrained the network to recover performance, which can be incorporated into deep compression channels through pruning, quantization, and Huffman Coding. Reference [13] proposed a framework for systematic weight pruning of DNNs using the alternating direction method of multipliers (ADMM). First, they formulated the DNN weight pruning problem as a nonconvex optimization problem with combinatorial constraints specifying sparsity requirements, which was then subjected to systematic weight pruning using the ADMM framework. The original nonconvex optimization problem was decomposed into two subproblems by using ADMM and was solved iteratively. One of the subproblems can be solved by stochastic gradient descent and the other can be solved by analytical methods. Niu et al. [14] advanced the state of the art by introducing a new two-dimensional space, namely fine-grained pruning patterns in coarse-grained structures, to re-reveal an unknown point in the design space. Due to the higher accuracy of fine-grained pruning patterns, their unique perspective is to use the compiler to recapture and guarantee high hardware efficiency. Liu et.al [15] proposed a frequency domain dynamic pruning scheme to exploit the spatial association of the frequency domain. Dynamic pruning of the frequency domain coefficient was performed on each iteration of the frequency domain coefficients and for differential pruning based on their different importance to the accuracy.

Filter pruning is trimmed at the level of the filter, channels, and even a single layer. Since the original convolutional structure is still preserved, no specialized hardware is required to achieve these effects. Similar to weight trimming, He et al. [16] proposed Learning Filter Pruning Criteria (LFPC) to address the above problems. Specifically, they developed a distinguishable pruning criteria sampler. This sampler was learnable and optimized by the validation loss of the pruning network obtained from the sampled criteria. In this way, they could adaptively select the appropriate pruning criteria for different functional layers. In addition, when evaluating the sampled criteria, LFPC comprehensively considers the contribution of all the layers at the same time. In reference [17], a novel greedy approach called cluster pruning was proposed, which provides a structured way of removing filters in a CNN by considering the importance of filters and the underlying hardware architecture. Zuo et al. [18] proposed a method of filter pruning without damaging the network capacity. They paid more attention to the damage by filter pruning to the model capacity. Zuo et al. [18] optimized the scaling factor γ in the BN layer as a channel-selection indicator to determine which channel was unimportant and capable of being pruned. Luo et al. [19] introduced ThiNet, which formally establishes filter pruning as an optimization problem, and reveals that we need to trim the filter based on the statistics computed from its next layer, and not based on the current layer. Likewise, Yu et al. [20] optimized the reconstruction error of the final response layer and propagated an “importance score” for each channel. He et al. [21] first proposed to perform model compression using AutoML and provided a strategy for model compression using reinforcement learning. Zhao et al. [22] performed the pruning operation by clustering the filters. Lin et al. [23] proposed an efficient structured pruning method for jointly pruning filters and other structures in an end-to-end manner. Specifically, the authors introduced a soft mask that extends the output of these structures by defining a new sparsity regularization objective function to adjust the output of the baseline and network with the output of this mask.

Among them, there is also a more detailed pruning scale in the filter pruning method, namely pruning by group. Xie et al. [24] constructed the Extending Filter Group (EFG) by conducting thorough investigations of the underlying constraints between every two successive layers. The penalty in terms of EFG addresses the training process on the filters of the current layer and the channels in the following layer, which is called synchronous reinforcement. Thus, it provides an alternative way to induce a model with ideal sparsity, especially in the case of complex datasets. Moreover, they presented Noise Filter Recognition Mechanism (NFRM) to improve model accuracy. Liu et al. [25] proposed a layer-grouping algorithm to find coupled channels automatically. Then, a unified metric based on Fisher information was derived to evaluate the importance of a single channel and coupled channels. A dynamic regularization approach to improve group pruning was proposed by Wang et al. [26]. However, trimming by group can remove weights in the same location in all filters in a level. Since the invalid location of each filter may vary, trimming by group may result in a neural network loss of groups accompanied by improved prediction accuracy. The different types of pruning patterns are illustrated in the following Figure 2.

In the method of neural-network-model-compression acceleration, we can also compress the number of parameters, and achieve the purpose of squeezing the neural network [27]. However, the compression of the parameters during the quantification process leads to a decrease in accuracy. In quantification methods, binary triple quantification is more common, where binary quantification is generally used to convert weights and activation values to 0/1 or 1/−1. There are two ways to binarize all the weights and the activation value of each layer. The first is the symbolic function, namely f(x)=1 if x>0, f(x)=−1 if x<0. The other is assigned with a certain probability, similar to dropout. Binarized Neural Networks is the second binarization method when activating the function, and the rest is assigned by the symbolic function. Triple quantization has more first-order weight than binary quantization. It is common practice to quantify weights to order 3, including 1, −1, 0. The specific approach is to minimize the Euclidean distance between the full precision weights and the quantified weights, using the following formula for mapping:(1)$$W_{i}^{t}=f_{t}\left(w_{i}|\Delta \right)=\{\begin{array}{c} +1,\;\;\;\;\;\;if\;W_{i}>\Delta \\ \;\;\;\;\;\;\;0,\;\;\;\;\;\;if\;\left|W_{i}\right|\leq \Delta \\ -1,\;\;\;\;\;\;if\;W_{i}<\Delta \end{array},\;\Delta =\frac{0.7}{n}\sum_{i=1}^{n}\left|W_{i}\right|$$
where n is the number of convolutional kernels, and i represents the convolutional kernel corresponding to the weights. However, although this kind of quantification method of converting high-precision weights to 2-value and 3-value has a very high compression ratio, the accuracy-reduction problem is still relatively serious for some network models. Accordingly, other quantification methods that have little impact on accuracy need to be selected.

## 3. The Proposed Pruning Algorithm Based on Strip Filter

### 3.1. Filter-Strip-Based Pruning

To obtain more refined filter pruning results, the practice of splitting the entire filter into filter strips was used here, as shown in Figure 3.

After breaking the filter into the filter strip, the filter strips in each filter were importance-evaluated by the $\ell_{1}$-norm of the weights in the filter strip, the sum of the weights, and the standard deviation. High-importance filter strips were retained, and other low-importance filter strips were pruned. Suppose the weight W of the 1st convolutional layer is $R^{N*C*K*K}$, where N is the number of filters, C the size of the channel and K the size of the convolutional kernel. Then, the size of the filter strip in this layer is $R^{N*K*K}$. A matrix initialization was first performed for the filter strip of each layer. During training, we multiplied the filter weights by the filter strip. Mathematically, the loss representation is shown:(2)$$L=\underset{\left(x,y\right)}{\sum }loss\left(f\left(x,W\cdot I\right),y\right)$$
where W represents the filter strip, and · indicates point product. The process of the forward transmission of the *I* is:(3)$$X_{n,h,w}^{l+1}=\overset{C}{\underset{c}{\sum }}\overset{K}{\underset{i}{\sum }}\overset{K}{\underset{j}{\sum }}I_{n,i,j}^{l}\times W_{n,c,i,j}^{l}\times X_{n,h+i-\frac{k+1}{2},w+j-\frac{w+1}{2}}^{l}$$
and the gradients regarding *W* and *I* are:(4)$$grad\left(W_{n,c,i,j}^{l}\right)=I_{n,i,j}^{l}\times \overset{M_{H}}{\underset{h}{\sum }}\overset{M_{W}}{\underset{w}{\sum }}\frac{\partial L}{\partial X_{n,h,w}^{l+1}}\times X_{c,h+i-\frac{K+1}{2},w+j-\frac{K+1}{2}}^{l}$$
(5)$$grad\left(I_{n,i,j}^{l}\right)=\overset{C}{\underset{c}{\sum }}\left(W_{n,c,i,j}^{l}\times \overset{M_{H}}{\underset{h}{\sum }}\overset{M_{W}}{\underset{w}{\sum }}\frac{\partial L}{\partial X_{n,h,w}^{l+1}}\times X_{c,h+i-\frac{K+1}{2},w+j-\frac{K+1}{2}}^{l}\right)$$

Among these, $M_{H}$, C represent the height and width of the feature graph when p<1 or $p>M_{H}$ or q<1 or $q>M_{H}$, $X_{c,p,q}^{l}=0$. Starting with Equation (2), the convolutional layer weights and filters were jointly optimized. Because W is only used here during evaluation, there is no additional cost to the network when applying inference. To illustrate the importance of the filter groups, an experiment was performed where the filter weights were fixed and only the filter-strip groups were trained during training. The test results are detailed in Table 1.

As can be seen from the data in the table, the identification accuracy of the whole model for training only the filter strips and not training the filter-related weights was still not low. In Figure 3, not all filter strips had the same effect on network accuracy recognition. To build compact and highly trimmed networks, the filtering filters require sparsity. When some weights in the filter strip approach 0, the corresponding filter strip (FS) can be trimmed. When training the network with FS, we regularized the FS to make it sparse:(6)$$L=\underset{\left(x,y\right)}{\sum }loss\left(f\left(x,W\cdot I\right),y\right)+\alpha g\left(I\right)$$

When α controls the size of the regularization, g(I) denotes the $\ell_{1}$-norm penalty on I, which is used in many pruning methods [16,18]. Specifically, g(I) is defined as the Equation (7):(7)$$g\left(I\right)=\overset{L}{\underset{l=1}{\sum }}g\left(I^{l}\right)=\overset{L}{\underset{l=1}{\sum }}\left(\overset{N}{\underset{n=1}{\sum }}\overset{K}{\underset{i=1}{\sum }}\overset{K}{\underset{j=1}{\sum }}\left|I_{n,i,j}^{l}\right|\right)$$

From Equation (6), the filter strip learns the best results of combining the entire-filter matching weights by training. Consequently, to guide the effective pruning, the method sets a threshold δ, which is not updated during the training process for corresponding values less than δ in FS and can be pruned later. When performing inference on the trimmed network, with the filter as a whole, we cannot directly convolute using the filter, because the input feature graph is corrupted. We need to independently use each band to perform convolution and summarize the feature maps generated by each band. Mathematically, the convolution process in the filter-strip pruning is written as:(8)$$X_{n,h,w}^{l+1}=\overset{K}{\underset{i}{\sum }}\overset{K}{\underset{j}{\sum }}\left(\overset{C}{\underset{c}{\sum }}W_{n,c,i,j}^{l}\times X_{n,h-i+\frac{K+1}{2},w-j+\frac{K+1}{2}}^{l}\right)$$
where $X_{n,h,w}^{l+1}$ is a point of the feature graph in layer l+1. From Equation (8) onward, the filter strip will only modify the order of calculations during traditional convolution, so it does not add computational quantities (Flops) to the network. The entire-filter pruning, to the mode of recombination through the filter strip, is shown in the Figure 4.

Because each stripe has its own position in the filter, the index of all bands are recorded. However, it has little computational cost compared to the entire network parameters. Assuming the weight of the layer-1 convolutional layer is N×C×K×K, for the entire filter, we need to record N×K×K indexes. We reduced the index of the weight pruning by C fold compared to individual weight pruning recording the N×C×K×K index. To make a fair comparison with traditional filter-pruning-based methods, we added the number of indexes when calculating the number of network parameters. The pruning training procedure is shown in the Figure 5.

### 3.2. Quantification of Data

The parameters used by convolutional neural network models for image recognition are generally 32-bit floating-point types. The calculation cost of the high-precision floating-point number type is greater than that of the integer data, so we can choose a method of quantifying the remaining parameters to greatly compress the number of remaining weight parameters and reduce the computational cost. Implementing quantization with neural networks requires converting convolution, matrix multiplication, activation functions, pooling, and splicing into equivalent 8-bit integer operations, and then adding quantization operations before and after the operations, which convert the input from a floating point to an 8-bit integer, and then convert the output from an 8-bit integer back to a floating point. Doing so minimizes the loss of precision from quantization.

At the same time, the incremental network quantification method was used here, by grouping the weights, which were quantified by groups, to retrain the three parts to complete the quantification. Doing so with the command can quantify the floating-point network as a low network. After quantization on the ResNet18 network, the 8-bit network was capable of exceeding the floating-point weights. Similar to network pruning, quantification is also the gradual removal of unimportant weights from the already trained grids, and the final result will not be significantly reduced, so the weights are of different importance. However, previous methods did not take this into account, but simultaneously converted high-precision floating-point numbers into low-precision values, therefore the importance of changing network weights is important to reducing the loss of the quantified network. Quantification through neural networks requires convolution calculations, matrix multiplication, activation function calculations, pooling layer calculations, and splicing operations to be converted to low-precision data for computation. We know that when low-precision data is computed in a computer, the computational cost is much lower than high-precision data.

After increasing the packet quantization process, the results of the filter-strip recombination can be taken as the result basis of the quantitative grouping on the basis of the previous pruning reorganization, where the quantization step can be interspersed into each convolutional calculation. That is, after the 32-bit floating-point-type input convolution layer, the 8-bit is quantified according to the filter group, then the convolution calculation yields the results, and the results are converted to the 32-bit floating-point-type output. This reduces the computational cost as much as possible and accelerates the computational efficiency of the model. Quantifying the data according to the results after the grouping allows some data that do not participate in the calculation to be directly output and simplifies the calculation process. The detailed procedure of the quantification is shown in Figure 6.

## 4. Experimental Analysis

The dataset used in the experiment was that of the training models used by CIFAR-10, which had VGG16 and ResNet56. The GPU configuration used was that of the GTX1070, and the CPU configuration was E5-2630. The whole experimental procedure was to first train the original model to converge using the dataset CIFAR-10, which is a public dataset. All of the types of comparisons are shown in this section.

### 4.1. Comparison between the Proposed Algorithm and Baseline Algorithm

Firstly, the model baseline data were obtained, and then they trained the model to different degrees according to different set thresholds. The model after the pruning training was processed for data-quantification training. Finally, a fully compressed model was obtained. The training baseline was set by using the CIFAR-10 as a dataset, the model was trained for 160 rounds, and the learning rate was set to 0.1. Plots of the training rounds and accuracy relations for pruning ratios at different thresholds for different models were selected. Experimental comparisons were made using different thresholds in the VGG16 and ResNet56 models, respectively. The first use was the contrast of pre- versus post-pruning accuracy, performed by the VGG16 model, where the threshold used was 0.01 and the results are shown in Figure 7.

It can be seen from the figure that the whole-model accuracy did not decrease in accuracy due to part of the pruning. In the curve section of the end of the right picture, the highest accuracy difference is no more than 1% before and after the whole pruning.

### 4.2. Comparison Based on Different Threshold between VGG16 and ResNet56

In order to obtain the maximum pruning rate and the highest accuracy based on the maximum pruning rate, the pruned models were trained and tested for accuracy at different thresholds, respectively. Based on the experimental results, it was shown that the overall computation and parameters of the model after pruning were substantially reduced, and there was no substantial decrease in the accuracy test results under the same dataset. The pruning method effectively compressed the computation of the model. The data of the experimental results are shown in the following Table 2.

Then, the ResNet56 model was selected, and the results of the rise curve of the ResNet56 accuracy and the loss drop curve before and after pruning during 160 rounds of training are shown in the following Figure 8.

Although throughout the whole training process, the accuracy under different pruning thresholds showed a certain decline, the overall accuracy curve direction was consistent. In the Figure 9, it can be seen that the highest change in the accuracy during training accompanied the increase in the pruning threshold.

In Figure 9, when the threshold is 0, pruning is not performed. The whole model was the least accurate at the threshold adjusted to 0.05, but the simultaneous accuracy loss was also controlled at around 0.01. At the same time, the threshold of 0.04 was similar to the threshold of 0.03. Both the number of parameters of the model and the computation amount somewhat decreased. The values corresponding to the respective pruning thresholds are shown in the following Table 3.

At the same time, we compared the accuracy effect of the filter pruning method based on the ResNet56 model, where the threshold selection of the filter pruning was based on the pruning rate. The experimental results are shown in the Figure 10.

### 4.3. Comparison Based on Different Methods

After filter-strip pruning, we quantified the weight data in the network, quantified the weights of the original data type as float32 to int8 type int8, and then restored them to the original data-type output. The final network model accuracy loss was less compared with the original model, but the computing cost and parameters fell more. In the method used to contrast, PF [28] is the method of filter pruning, and the evaluation standard uses the $\ell_{1}$-norm value of the parameters, the standard deviation, and the value. The SFP [29] method adopts the dynamic pruning method, mainly relying on the norm values of the weights, retaining the sparse bias of the BN layer. GAL [30] is the filter pruning of generated adversarial network learning. The results are shown in the Table 4.

The model-parameter-compression rates of different methods of different models can be seen from the table. The effect of this method was better than the other methods, and the accuracy loss was also the minimum.

At the same time, we can note that reference [20] and reference [31] also showed good improvements. In this regard, many people have achieved good results. For example, reference [31] proposed a new regularizer on scaling factors, namely a polarization regularizer, to achieve the state-of-the-art result. Reference [32] describes two variations of our method using the first- and second-order Taylor expansions to approximate a filter’s contribution. Wu [33] proposed adversarial neuron pruning (ANP), which prunes some sensitive neurons to purify the injected backdoor and improve the robustness of the model. The next step can be to try to analyze the causality in the connections between neurons to further improve the model.

## 5. Case Study

### 5.1. Application Background and Environment

The research content of object detection recognition based on convolutional neural networks has become increasingly mature over time [25]. The current computing power of mobile devices is also constantly improving, so the task requirements based on the target detection and recognition of mobile devices also follow [34]. By using an autonomous panorama-vision-based inspection system, the limitations of the human cost and safety factors of previously time-consuming tasks have been overcome [35]. Liu [36] introduced coefficient matrices regularized by a variety of regularization terms to locate important kernel positions. On the one hand, the cameras carried by mobile phones or other mobile terminals can obtain higher and higher quality images, and even take professional photos; on the other hand, the recognition speed of algorithms and models is becoming higher and more accurate, and can run in real time on various mobile terminals, such as mobile phones, tablets or drones. Based on this, we implemented a target recognition and detection system on Android by combining the previously proposed filter-bar pruning method and packet data quantization method. 

### 5.2. System Framework

The whole algorithm process of the target-identification-detection system is shown in Figure 11, and the compression of the model is performed according to the separation filter-strip pruning and packet quantization method given above. In the model-training phase, we still used the parametric model to improve the detection accuracy of the model. At the same time, it was combined with dropout and other methods to prevent the model from over-fitting. Filter pruning was then combined with our proposed filter-strip pruning method, and the packet data after the filter-strip reorganization were retained as the grouping basis for data quantization. Finally, the model was quantified at 8-bit using the tool chain given in the NCNN and finally compiled to the ARM platform.

### 5.3. System Deployment and Testing

The target-detection-and-identification system mainly uses the VGG16 model for identification and detection [19,20,37]. This paper focuses on identification using the trained and converged VGG16 model, and compares the detection results with the VGG16 model that underwent pruning and quantization methods. To better show the object-recognition effect deployed on mobile phones, this paper presents the VGG16 model combined with the FASTER RCNN algorithm to identify the identified targets using borders [21]. The main purpose was to select the target box in the image using methods such as border regression, but the main model used for identification is still the VGG16. In pruning based on the above method, the accuracy of the VGG16 model after training at different selected thresholds is shown in the Figure 12.

Among them, there were 160 training rounds, and it can be seen that under the VGG model combined with the data shown in Figure 12 and Table 2, the accuracy of 0.02 only decreased by about 0.001 compared to the threshold of 0.01, but the parameter number (Params) decreased more at the 0.01 threshold compared with the calculated amount (flops). Thus, the pruning threshold chosen here was 0.02. Finally, pruning training was performed, combined with group quantification during the later inferred computation.

However, because generated model files by pytorch were used here, but the NCNN does not support direct pytorch models directly into the software framework, the pytorch model was converted to ONNX, ONNX (Open Neural Network Exchange), which is an open-source file format designed for deep-learning models that allows model files generated by different AI frameworks to store model data in the same format in order to realize the purpose of migration in different frameworks [23,38]. The NCNN is a neural network computing framework developed by the Tencent mobile research team, with multiple built-in optimization frameworks. The NCNN is also used as the framework for deployment to the mobile terminal for forward computing. The entire model takes Android11 as the software running environment, and ARMx86 is the hardware running environment. Both target-recognition modes were combined with CPU acceleration calculation and GPU acceleration calculation [39].

The whole mobile terminal identification system construction process is as follows: in the first step, the system obtains the image data through the mobile-phone camera or system files. In the second step, the image data are sent into the model for extracting features, computing features, feature classification and other calculation processes. In the third step, the identified target and the identification results are identified in the image. The identification process framework for the entire system is shown in Figure 13.

After the system is constructed based on the above method, the interface and recognition effect of the whole software are shown in Figure 14.

At the same time, the system obtained the time of the identification result by image inference by CPU, and the time of the identification result by GPU inference. Two models have to infer time by comparing the average of 10 recognition times of the same recognition image. The results are shown in Table 5.

## 6. Conclusions

The main method used in this paper is a combination of two compression models: filter-strip pruning and data quantification. The filters in the neural network are firstly re-refined into strip-like filter strips. Second, the filter strips are evaluated to redetermine the partial importance of the filters, the unimportant filter strips are cut off and the remaining filter strips are reorganized. Finally, the training of the reorganized neural network inserts the quantitative processing and fine-tunes the network to further compress the computational amount of the neural network. The double-layer compression model is more suitable for mobile and embedded devices. Compared with some other traditional methods, the model size compression, accuracy, and inference time are somewhat improved. However, the method still has some shortcomings. There is still room for improvement in the selection of kernels to be pruned, and the pruning model can be further optimized. Based on the comparison results presented in Table 4, we can analyze the causality in the connections between the neurons and further optimize the model by manipulating finer structures. As for our current progress, experiments to analyze the causality in the connections between neurons will be the next step of our research.

## Figures and Tables

**Figure 1 sensors-22-05623-f001:**
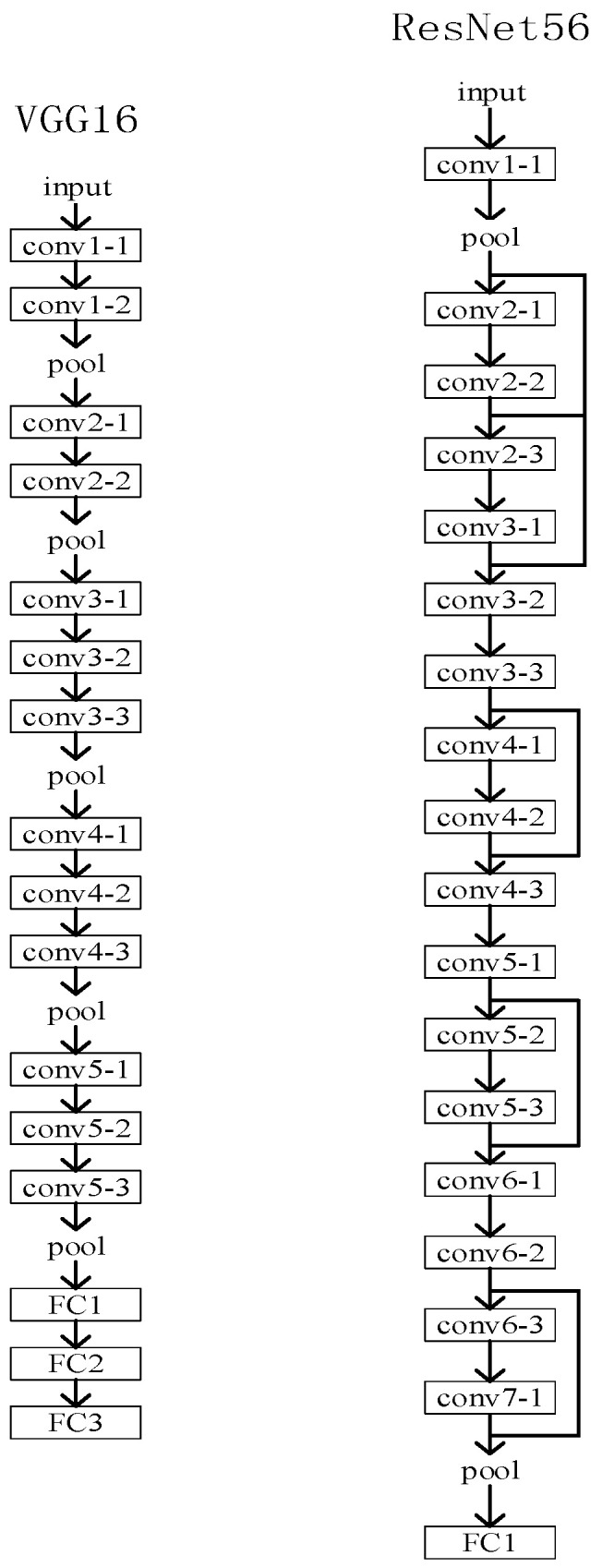
VGG16 and ResNet56 structural plots.

**Figure 2 sensors-22-05623-f002:**
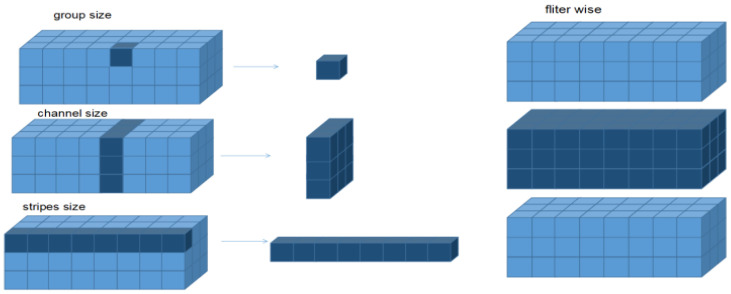
Filter pruning of different sizes.

**Figure 3 sensors-22-05623-f003:**
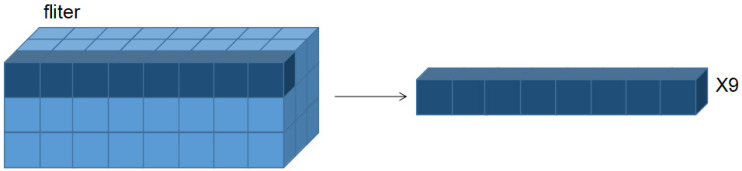
Filter decomposition.

**Figure 4 sensors-22-05623-f004:**
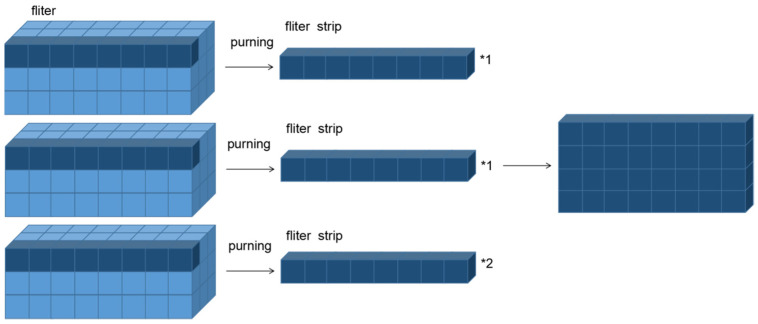
Filter-strip pruning and recombination.

**Figure 5 sensors-22-05623-f005:**
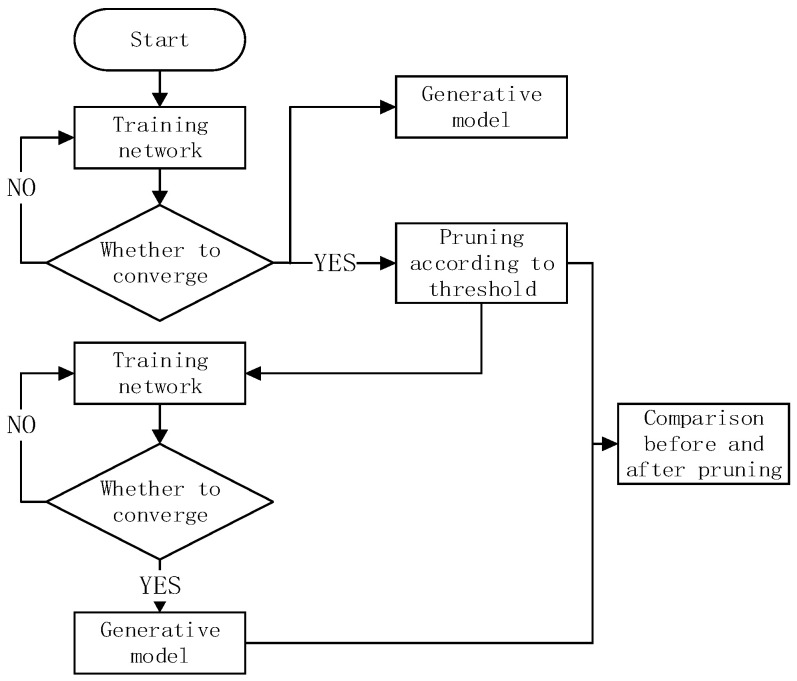
Searing training process.

**Figure 6 sensors-22-05623-f006:**
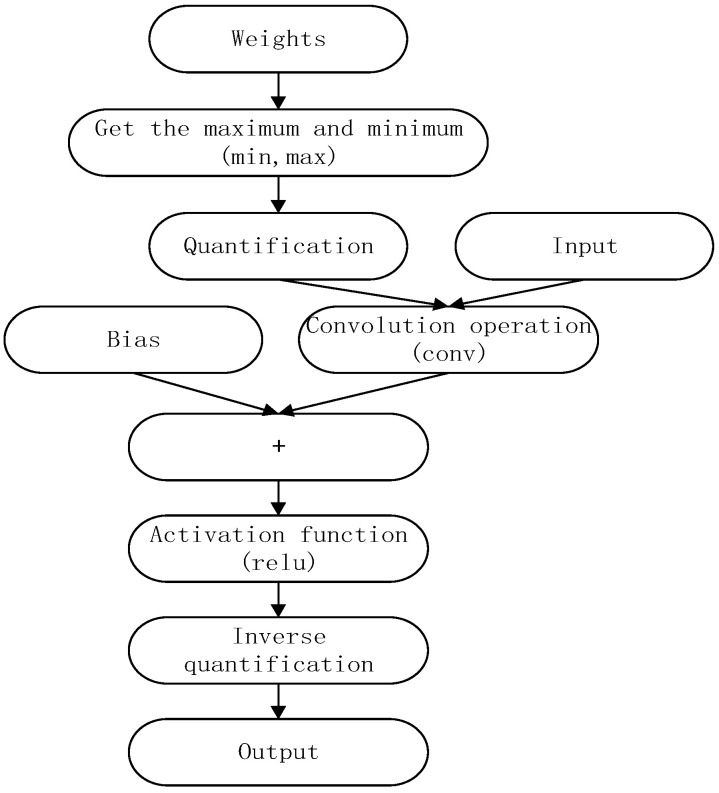
Quantifying the data process.

**Figure 7 sensors-22-05623-f007:**
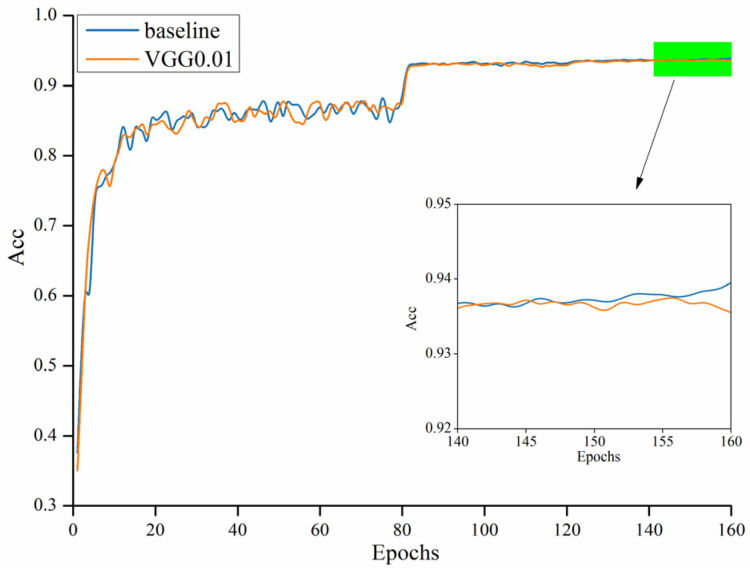
The comparison of accuracy before and after pruning of the VGG16 model.

**Figure 8 sensors-22-05623-f008:**
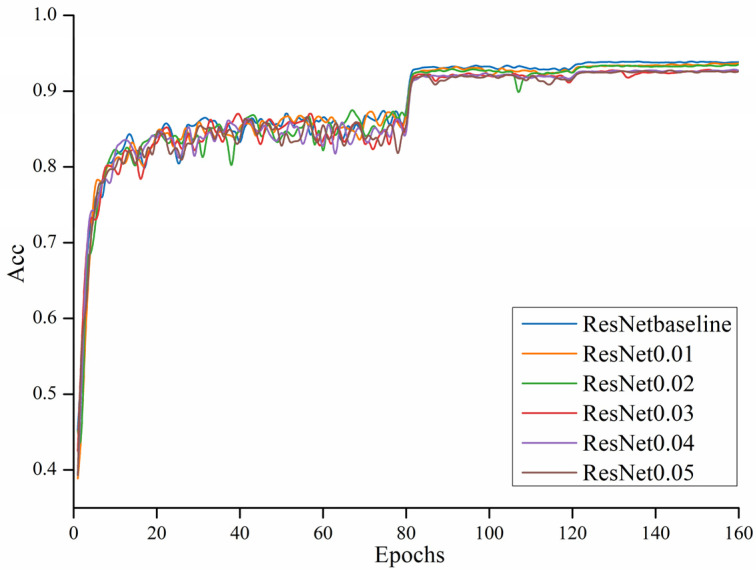
Accuracy comparison of the ResNet56 model.

**Figure 9 sensors-22-05623-f009:**
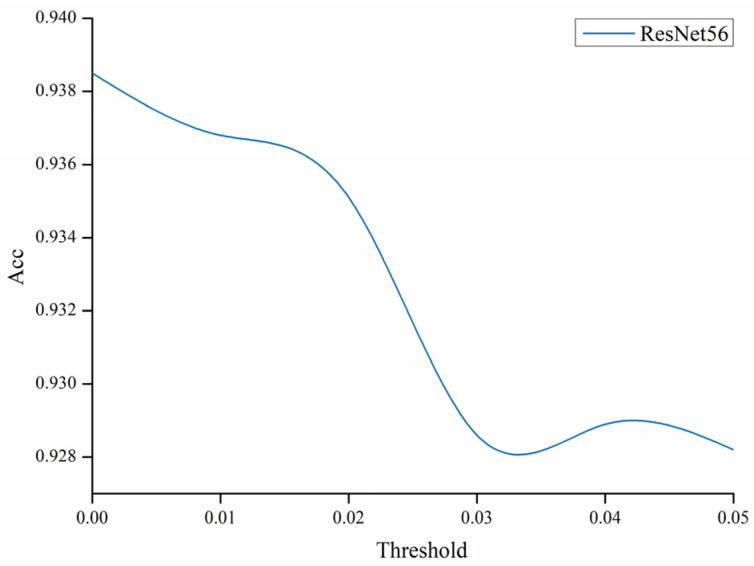
Changes in accuracy at different thresholds.

**Figure 10 sensors-22-05623-f010:**
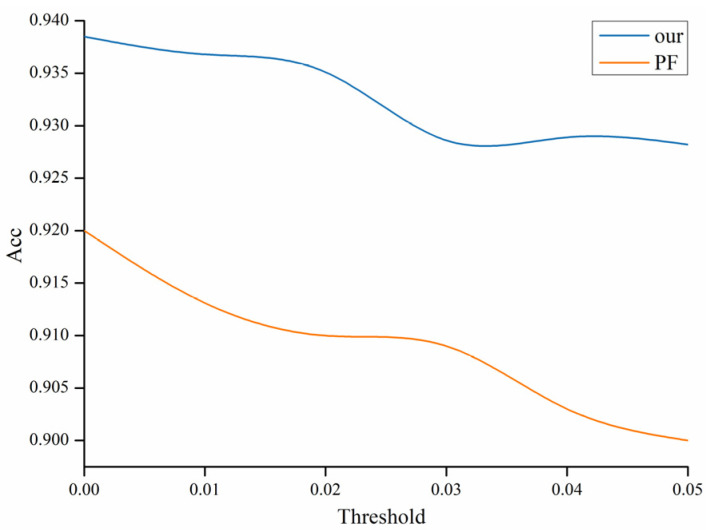
The comparison of filter pruning and filter-strip pruning.

**Figure 11 sensors-22-05623-f011:**
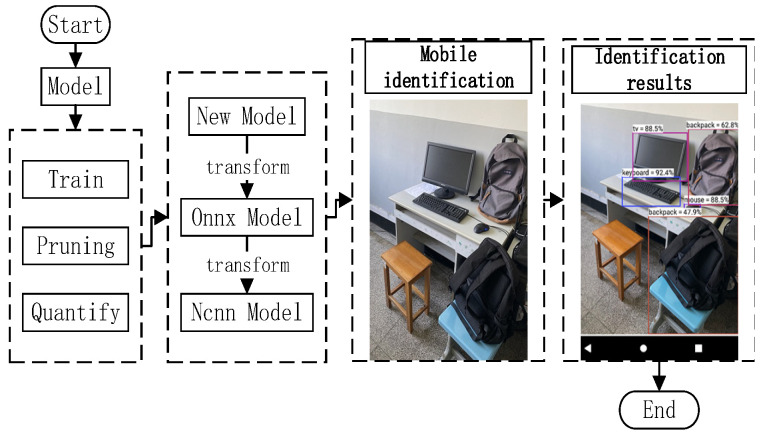
The algorithm process of the target-detection-and-identification system.

**Figure 12 sensors-22-05623-f012:**
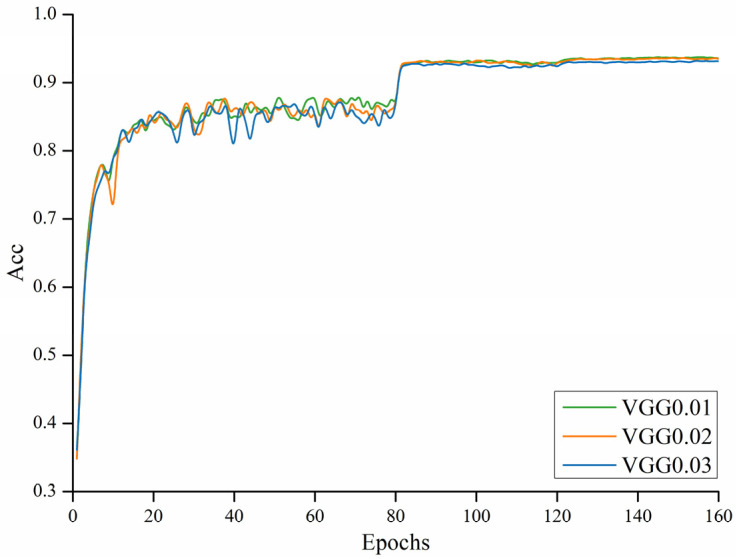
Training accuracy of the VGG model at different thresholds.

**Figure 13 sensors-22-05623-f013:**
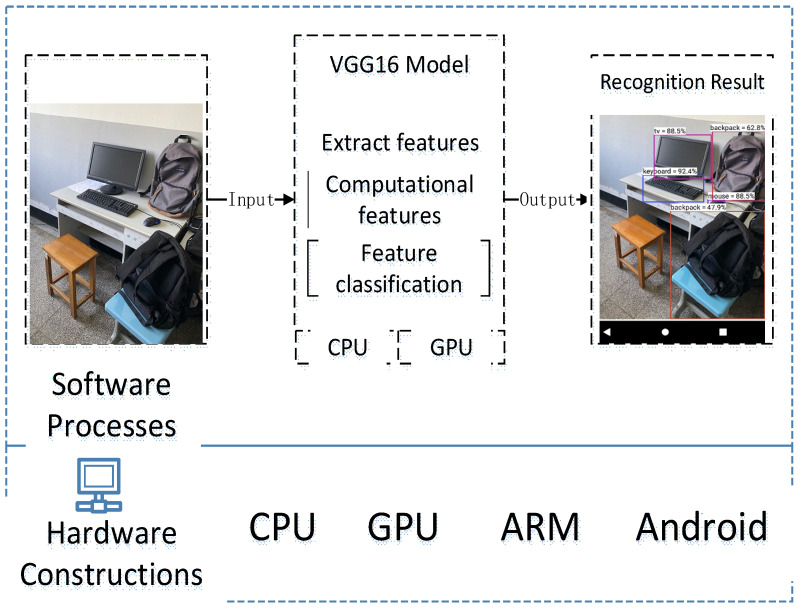
System construction process and environment.

**Figure 14 sensors-22-05623-f014:**
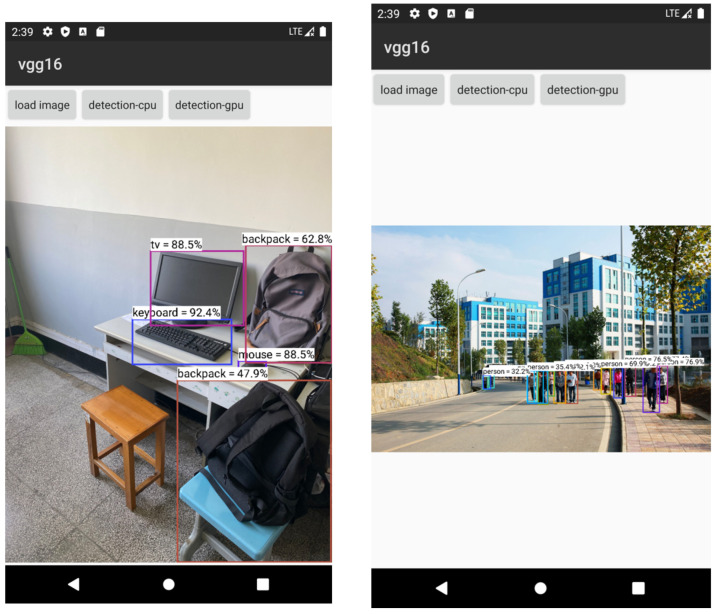
Software interface and recognition effect.

**Table 1 sensors-22-05623-t001:** Learn only the test accuracy for each network of the filter strip.

Data Set	Model	Accuracy
CIFAR-10	VGG16	79.8
	ResNet56	83.8
	MoblieNetV2	83.5

**Table 2 sensors-22-05623-t002:** Pruning results of the different thresholds of the VGG16 model.

Threshold (VGG16)	Accuracy	Flops (M)	Params (M)
0 (baseline)	0.9417	627	14.2
0.01	0.9377	281.09	2.38
0.02	0.9363	228.16	1.65
0.03	0.9322	185.8	1.1
0.04	0.9351	163.06	0.82
0.05	0.8714	233.3	1.2

**Table 3 sensors-22-05623-t003:** Pruning results for the different thresholds of the ResNet56 model.

Threshold (ResNet56)	Accuracy	Flops (M)	Params (M)
0 (baseline)	0.9385	251	0.86
0.01	0.9368	114.37	0.46
0.02	0.9351	93.08	0.4
0.03	0.9286	79.5	0.33
0.04	0.9289	62.42	0.3
0.05	0.9282	57	0.23

**Table 4 sensors-22-05623-t004:** Effect comparison of the pruning method.

Backbone	Metrics	Accuracy	Flops (M)	Params (M)
ResNet56	Baseline	0.9385	251	0.86
	PF	0.9131	90.9	0.38
	SFP	0.931	107	0.41
	[20]	0.9085	141.5	0.494
	[31]	0.9383	133.03	-
	Our	0.9351	93.08	0.4
VGG16	Baseline	0.9417	627	14.2
	PF	0.9310	412	5.112
	SFP	0.9208	226	5.12
	GAL	0.9278	378.7	3.2
	[31]	0.9392	288.42	-
	Our	0.9322	185.8	1.1

**Table 5 sensors-22-05623-t005:** Comparison of the inferred time before and after model pruning.

Model	CPU Time	GPU Time
VGG16	631.37 ms	279.39 ms
VGG16 pruning	529.37 ms	103.28 ms

## Data Availability

Not applicable.

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
