# Peer review of "A Novel Deep-Learning Model Compression Based on Filter-Stripe Group Pruning and Its IoT Application"

_sensors, 2022, doi:10.3390/s22155623_

Round 1

Reviewer 1 Report

In this work, the authors proposed a pruning strategy for DNN, as well as the case study for IoTs. Lot's of researches about it. From the overall view, I would not sign my report from the below considerations:

(1) The authors can find the below attach files with my comments and highlights in the manuscript (not fully). The writing must be well organized from the begin to end. 

(2) Figures 7, 8, 9 and 10, are worse. It's far from being published. 

(3) FS and NCNN, should be explained... VGG, ResNet in the article should keep the same at the context.

(4) I would like the author to cite the official published paper, rather than in arxiv, in Ref. 1, 7, 20;

(5) The reference format should be careful in writing.

(6)  The dataset used in the article should clear its data source.

I suggest the authors to take the above comments in serious. The article can be resubmit until you are satisfied with it firstly, from its writing, figures, references and so on.

Author Response

Dear Reviewer:

Thank you for your letter and for the reviewers’ comments concerning our manuscript entitled “A Novel Deep Learning Model Compression Based on Filter-Stripe Group Pruning and Its IoT Application” (ID: sensors-1797044). Those comments are all valuable and very helpful for revising and improving our paper, as well as the important guiding significance to our research. We have studied comments carefully and have made correction which we hope to meet with approval. Revised portions are marked in yellow in the paper. The main corrections in the paper and the responds to the reviewer’s comments are as flowing:

Responds to the reviewer’s comments:

Reviewer #1

Point #1 The authors can find the below attach files with my comments and highlights in the manuscript (not fully). The writing must be well organized from the begin to end.

Answer:Thank you for your comments. We have made changes in the appropriate places and marked them in yellow. Regarding Figure 14, if it is cropped, it may not be sufficient to reflect that this application environment is a mobile device.

Point #2 Figures 7, 8, 9 and 10, are worse. It's far from being published.

Answer:Thank you for your advice. We have rechecked all the images and redrawn the bad ones.

Point #3 FS and NCNN, should be explained... VGG, ResNet in the article should keep the same at the context.

Answer:Thanks for valuable comments. FS is short for filter strip in the article, and we have added it in the corresponding place. NCNN is a neural network computing framework developed by Tencent for mobile research team. There is a detailed explanation of it in section 5.3. And we have corrected the irregular writing style and standardized the names of VGG and ResNet.

Point #4 I would like the author to cite the official published paper, rather than in arxiv, in Ref. 1, 7, 20;

Answer:Thank you for your advice. We have checked the references and corrected the irregular citations.

Point #5 The reference format should be careful in writing.

Answer:Thank you for your comments. We have corrected the irregular citations.

Point #6 The dataset used in the article should clear its data source.

Answer:Thank you for your advice. We use a public dataset, which we have added a note to in the text.

Reviewer 2 Report

In this article, the authors presented a strategy for refining pruning quantification and weights based on neural network filters. Experimental results showed that the proposed method can significantly reduce the computation of neural networks and compressed the number of parameters in the model.

However, there are several improvements that need to make before publication.

1.     Line 4, the authors' name seems not in the correct format, the “and” may not be in the right place.

2.     Line 7, the X.D.T is not the same font as the others. Similar mistakes as in Line 9 and Line 11

3.     Line 20, resnet56 should be Resnet56 as in capital format.

4.     Line 54, Figure 1, the VGG-16 only showed the input arrow without any other arrow to demonstrate the data flow. The quality of Resnet56 structure should be improved. It seems to be drawn by hand.

5.     Line 263, Figure 7 is not very clear. Do not use similar colors in the same image.

6.     Line 280, Figure 8, what is the number above the Epoch?

7.     In reference, there should be more recent works to demonstrate the IoT application, such as:

a.     Luo, Cai, Leijian Yu, Jiaxing Yan, Zhongwei Li, Peng Ren, Xiao Bai, Erfu Yang, and Yonghong Liu. "Autonomous detection of damage to multiple steel surfaces from 360 panoramas using deep neural networks." ComputerAided Civil and Infrastructure Engineering 36, no. 12 (2021): 1585-1599.

b.     Liu, Guangzhe, Ke Zhang, and Meibo Lv. "SOKS: Automatic Searching of the Optimal Kernel Shapes for Stripe-Wise Network Pruning." IEEE Transactions on Neural Networks and Learning Systems (2022).

Author Response

Dear Reviewer:

Thank you for your letter and for the reviewers’ comments concerning our manuscript entitled “A Novel Deep Learning Model Compression Based on Filter-Stripe Group Pruning and Its IoT Application” (ID: sensors-1797044). Those comments are all valuable and very helpful for revising and improving our paper, as well as the important guiding significance to our research. We have studied comments carefully and have made correction which we hope to meet with approval. Revised portions are marked in yellow in the paper. The main corrections in the paper and the responds to the reviewer’s comments are as flowing:

Responds to the reviewer’s comments:

Reviewer #2

Point #1 Line 4, the authors' name seems not in the correct format, the “and” may not be in the right place.

Answer:Thank you for your advice. We have corrected the errors.

Point #2 Line 7, the X.D.T is not the same font as the others. Similar mistakes as in Line 9 and Line 11.

Answer:Thank you for your advice. We have corrected the formatting.

Point #3 Line 20, resnet56 should be Resnet56 as in capital format.

Answer:Thanks for valuable comments. We have standardized the names of ResNet.

Point #4 Line 54, Figure 1, the VGG-16 only showed the input arrow without any other arrow to demonstrate the data flow. The quality of Resnet56 structure should be improved. It seems to be drawn by hand.

Answer:Thank you for your advice. We have redrawn this image.

Point #5 Line 263, Figure 7 is not very clear. Do not use similar colors in the same image.

Answer:Thank you for your comments. We have redrawn this image to make it clearer.

Point #6 Line 280, Figure 8, what is the number above the Epoch?

Answer:Thank you for your advice. We have redrawn this image to make it clearer.

Point #6 In reference, there should be more recent works to demonstrate the IoT application, such as:

  1. Luo, Cai, Leijian Yu, Jiaxing Yan, Zhongwei Li, Peng Ren, Xiao Bai, Erfu Yang, and Yonghong Liu. "Autonomous detection of damage to multiple steel surfaces from 360 panoramas using deep neural networks." Computer‐Aided Civil and Infrastructure Engineering 36, no. 12 (2021): 1585-1599.
  2. Liu, Guangzhe, Ke Zhang, and Meibo Lv. "SOKS: Automatic Searching of the Optimal Kernel Shapes for Stripe-Wise Network Pruning." IEEE Transactions on Neural Networks and Learning Systems (2022).

Answer:Thank you for your comments. We have added more recent references.

Reviewer 3 Report

The contribution of this paper, in the scientific field of Machine Learning, in my opinion, is high rated, because of the importance in the weight recording positions, in order to estimate the results of the weight pruning procedure, in the predicted performance. An also important issue is at the analysis in the trade off, between the kernel's size reduction and the model's performance. A future challenge could be the question about how we can determine the optimal kernel size of each filter, without the pruning technique, a priori? Thank you.

Author Response

Dear Reviewer:

Thank you for your letter and for the reviewers’ comments concerning our manuscript entitled “A Novel Deep Learning Model Compression Based on Filter-Stripe Group Pruning and Its IoT Application” (ID: sensors-1797044). Those comments are all valuable and very helpful for revising and improving our paper, as well as the important guiding significance to our research. We have studied comments carefully and have made correction which we hope to meet with approval. Revised portions are marked in yellow in the paper. The main corrections in the paper and the responds to the reviewer’s comments are as flowing:

Responds to the reviewer’s comments:

Reviewer #3

Point #1 In the deep neural networks, models' architecture has high complexity, so it becomes necessary for pruning techniques, to be developed. Because of the stochasticity, it must be determined the kind of the distribution in the trade off, between the complexity and the accuracy. The paper could be imrpoved by analyzing the causality in the connections between the neurons.

Answer:Thanks for your valuable comments. In this paper, we focus on the filter strips of the filters, and have not studied the more detailed structure. In the process of comparing the experimental results, we also noticed that many people made some results in the direction of causality of the connections between neurons Based on our current progress, this will be one of our next steps, and we will work in this direction to further improve our model. Please give us a chance to complete the experiment in the future work。

Round 2

Reviewer 1 Report

Several comments:

(1) Figure 7, 8, 9, 10, 12 are worse resolution. acc-->Acc, threshold-->Threshold, etc., Please plot the lines with more readable style.

(2) I don't think the references are proper cited. 10/40 cites from conference/proceeding. Ref. 16 is not fully. Ref. 39 is not accessable. From the references, it shows that the authors are not familiar with this field, or can't catch the up-to-date papers. 

(3) I don't think the modification of conclusion is perfect. That would be better to move the implementation to the section of “discussion". 

I would like the authors to make a "MAJOR" revision from all the comments by reviewers. Otherwise, for reviewers, it's a waste of time to spend time to read, comment, to help the improvement for further publication. Thank you.

Author Response

Dear Reviewer:

Thank you for your letter and for the reviewers’ comments concerning our manuscript entitled “A Novel Deep Learning Model Compression Based on Filter-Stripe Group Pruning and Its IoT Application” (ID: sensors-1797044). Those comments are all valuable and very helpful for revising and improving our paper, as well as the important guiding significance to our research. We have studied comments carefully and have made correction which we hope to meet with approval. Revised portions are marked up using the “Track Changes” function. The main corrections in the paper and the responds to the reviewer’s comments are as flowing:

Responds to the reviewer’s comments:

Reviewer #1

Point #1 Figure 7, 8, 9, 10, 12 are worse resolution. acc-->Acc, threshold-->Threshold, etc., Please plot the lines with more readable style.

Answer:Thank you for your comments. We redrew the image to ensure the resolution of the image and modified the irregular writing in the figure.

Point #2 I don't think the references are proper cited. 10/40 cites from conference/proceeding. Ref. 16 is not fully. Ref. 39 is not accessable. From the references, it shows that the authors are not familiar with this field, or can't catch the up-to-date papers.

Answer:Thank you for your advice. We have modified the problematic citation format. About reference 39, we have checked all references. It can be searched by name or Doi in CNKI(cnki.net). it could be seen in reference [10], and reference [32] to [40].

Point #3 I don't think the modification of conclusion is perfect. That would be better to move the implementation to the section of “discussion".

Answer:Thanks for valuable comments. We restructured the article by placing part of the conclusion at the end of section 4.3.

Reviewer 2 Report

No further comment

Author Response

Dear Reviewer:

Thank you for your letter and for the reviewers’ comments concerning our manuscript entitled “A Novel Deep Learning Model Compression Based on Filter-Stripe Group Pruning and Its IoT Application” (ID: sensors-1797044). Those comments are all valuable and very helpful for revising and improving our paper, as well as the important guiding significance to our research. We have studied comments carefully and have made correction which we hope to meet with approval. Revised portions are marked up using the “Track Changes” function. The main corrections in the paper are labeled in yellow.

Round 3

Reviewer 1 Report

The reviewers spend time to help the paper for publication. However, I don't think the author show positive revisions, especially in  reference, as comment on 2nd round review.

Author Response

Dear Reviewer:

I'm sorry for my negligence, you are so kind to me, thank you for your letter and for the reviewers’ comments concerning our manuscript entitled “A Novel Deep Learning Model Compression Based on Filter-Stripe Group Pruning and Its IoT Application” (ID: sensors-1797044). Those comments are all valuable and very helpful for revising and improving our paper, as well as the important guiding significance to our research. We have studied comments carefully and have made correction which we hope to meet with approval. Revised portions are marked up using the “Track Changes” function. The main corrections in the paper and the responds to the reviewer’s comments are as flowing:

Responds to the reviewer’s comments:

Reviewer #1

Point #1 The reviewers spend time to help the paper for publication. However, I don't think the author show positive revisions, especially in reference, as comment on 2nd round review.

Answer: We highly thank reviewer for this suggestion.  We rearranged the references and their counter parts, replacing some updated papers. At the same time, we checked the grammar of the article and rewrote some expressions that we thought were inappropriate.   The Revised portions are labeled in yellow and using the ““Track Changes” function.”  Such as:

In Section 1, paragraph 2, we rewrote the content: Weight pruning is mainly implemented by pruning various weights in the network. When the value of a certain weight within the network layer is close to or equal to zero, it can be considered that the prediction performance will not be sacrificed by removing this weight, and hence pruning it to eventually form a sparse network.

The original text is as follows:

In Section 1, paragraph 3, the original text is as follows:

We rewrote it:

It further shows that the pruned model trained with random initialization also achieves high performance. Therefore, it is the structure of the network that matters, not the trained weights.

In Section 1, paragraph 4, the original text is as follows:

Zhang et al. [11]

We modified it as:

Zhang et al. [11] proposed a progressive, multi-step weight pruning framework, as well as network purification and unused path removal procedures to achieve higher pruning rates without loss of accuracy.

In paragraph 5, the revised portions are as follows: Based on this, a refined pruning and weight quantization method based on neural network filters is proposed.The filters in the neural network are firstly refined into stripe-like of filter strips.

In Section 2, paragraph 2, the revised portions and original text are as follows:

The literature [13] proposed a framework for systematic weight pruning of DNNs using the alternating direction method of multipliers (ADMM). First, they formulated the DNN weight pruning problem as a nonconvex optimization problem with combinatorial constraints specifying sparsity requirements, which was then subjected to systematic weight pruning using the ADMM framework. The original nonconvex optimization problem is decomposed into two subproblems by using ADMM and is solved iteratively. One of the subproblems can be solved by stochastic gradient descent and the other can be solved by analytical methods.Niu et al.[14] advanced the state-of-the-art by introducing a new two-dimensional space, namely fine-grained pruning patterns in coarse-grained structures, to re-reveal an unknown point in the design space. Due to the higher accuracy of fine-grained pruning patterns, their unique perspective is to use the compiler to recapture and guarantee high hardware efficiency.

In Section 2, paragraph 3, the revised portions and original text are as follows:

He et al. [16] proposed Learning Filter Pruning Criteria (LFPC) to address the above problems. Specifically, they develop a distinguishable pruning criteria sampler. This sampler is learnable and optimized by the validation loss of the pruning network obtained from the sampled criteria. In this way, they can adaptively select the appropriate pruning criteria for different functional layers. In addition, when evaluating the sampled criteria, LFPC comprehensively consider the contribution of all the layers at the same time. In the literature [17], a novel greedy approach called cluster pruning has been proposed, which provides a structured way of removing filters in a CNN by considering the importance of filters and the underlying hardware architecture. Zuo et al. [18] propose a method of filter pruning without damaging networks capacity. They pay more attention to the damage by filter pruning to model capacity.

In Section 2, paragraph 4, the revised portions and original text are as follows:

Xie et al. [24] construct the Extending Filter Group (EFG) by thorough investigations on underlying constraints between every two successive layers. The penalty in terms of EFG addresses train process on filters of the current layer and channels in the following layer, which called as synchronous reinforcement. Thus, it provides an alternative way to induce a model with ideal sparsity, especially in case of complex datasets. Moreover, they present Noise Filter Recognition Mechanism (NFRM) to improve model accuracy.Liu et al. [25] propose a layer grouping algorithm to find coupled channels automatically. Then we derive a unified metric based on Fisher information to evaluate the importance of a single channel and coupled channels.

In Section 3.2, paragraph 1, the revised portions and original text are as follows:

Implementing quantization with neural networks requires converting convolution, matrix multiplication, activation functions, pooling, and splicing into equivalent 8-bit integer operations, and then adding quantization operations before and after the operations, which convert the input from floating-point to 8-bit integer and then convert the output from 8-bit integer back to floating-point. Doing so minimizes the loss of precision from quantization.

In Section 3.2, paragraph 2, the revised portions and original text are as follows:

After quantization on the ResNet18 network, the 8-bit network was capable of exceeding the floating-point weights..

Quantification through neural networks requires convolution calculations, matrix multiplication, activation function calculations, pooling layer calculations, and splicing operations to be converted to low-precision data for computation.

In Section 4.2, paragraph 1, the revised portions and original text are as follows:

In order to obtain the maximum pruning rate and the highest accuracy based on the maximum pruning rate, the pruned models were trained and tested for accuracy at different thresholds, respectively.

In Section 4.3, paragraph 3, the revised portions and original text are as follows:

the literature [32] describes two variations of our method using the first and second order Taylor expansions to approximate a filter's contribution.

In Section 5.1, paragraph 1, the revised portions and original text are as follows:

Based on this, we implemented a target recognition and detection system on Android by combining the previously proposed filter bar pruning method and packet data quantization method.

In Section 5.1, paragraph 2, the revised portions and original text are as follows:

In the model training phase, we still use the parametric model to improve the detection accuracy of the model.

In section 5.2, paragraph 1, the revised portions and original text are as follows:

This paper focuses on their identification using the trained and converged VGG16 model, and also compares the detection results with the VGG16 model that has undergone pruning and quantization methods.

In section 5.2, paragraph 3, the revised portions and original text are as follows:

NCNN is a neural network computing framework developed by the Tencent mobile research team, with multiple built-in optimization frameworks.

In section 6, the revised portions and original text are as follows:

The filters in the neural network are firstly re-refined into strip-like filter strips.Second, the filter strips are evaluated to redetermine the partial importance of the filters, and the unimportant filter strips are cut off and the remaining filter strips are reorganized.

Finally, we updated references, such as reference [13], [14], [16], [17], [18], [24], [25], [27], [32], [37]. They are labeled in yellow, and they used the “Track Changes” function.
